# The Potential of Gamma Secretase as a Therapeutic Target for Cardiac Diseases

**DOI:** 10.3390/jpm11121294

**Published:** 2021-12-04

**Authors:** Sujoita Sen, Logan Hallee, Chi Keung Lam

**Affiliations:** 1Department of Biological Sciences, University of Delaware, Newark, DE 19716, USA; sujoita@udel.com; 2Department of Mathematical Sciences, University of Delaware, Newark, DE 19716, USA; lhallee@udel.com

**Keywords:** gamma-secretase, cardiac, disease, presenilin, proteolysis, signaling

## Abstract

Heart diseases are some of the most common and pressing threats to human health worldwide. The American Heart Association and the National Institute of Health jointly work to annually update data on cardiac diseases. In 2018, 126.9 million Americans were reported as having some form of cardiac disorder, with an estimated direct and indirect total cost of USD 363.4 billion. This necessitates developing therapeutic interventions for heart diseases to improve human life expectancy and economic relief. In this review, we look into gamma-secretase as a potential therapeutic target for cardiac diseases. Gamma-secretase, an aspartyl protease enzyme, is responsible for the cleavage and activation of a number of substrates that are relevant to normal cardiac development and function as found in mutation studies. Some of these substrates are involved in downstream signaling processes and crosstalk with pathways relevant to heart diseases. Most of the substrates and signaling events we explored were found to be potentially beneficial to maintain cardiac function in diseased conditions. This review presents an updated overview of the current knowledge on gamma-secretase processing of cardiac-relevant substrates and seeks to understand if the modulation of gamma-secretase activity would be beneficial to combat cardiac diseases.

## 1. Introduction

Cardiac diseases have been a major health issue around the globe. According to the American Heart Association approximately 6 million Americans had heart failure between 2015 and 2018 [1]. Hence, it is of vital importance to find new and effective therapeutic remedies to address cardiac diseases. In this regard, gamma-secretase could be a new, promising target for the intervention of heart disease. Gamma-secretase is an aspartyl protease complex involved in processing an extensive array of type 1 transmembrane proteins via intra-membrane cleavage. It has been studied extensively in the context of Alzheimer’s disease, as it is directly involved in the production of toxic amyloid-β from the amyloid precursor protein (APP).

Given that gamma-secretase can cleave a wide array of transmembrane proteins, it has been associated with a number of downstream processes [2]. For example, Notch proteins are prominent (and potentially cardioprotective) substrates of gamma-secretase. These proteins are linked to processes in cell growth and development, as well as cell and tissue morphogenesis in metazoans [3]. Prior studies have established the crucial role of Notch proteins in the development of the cardiac system and repair processes of post-cardiac injury [4]. When looking for substrates related to heart function, we hoped to find many that are potentially beneficial to target in cardiac diseases. Many drugs have been designed to inhibit and modulate gamma-secretase activity as an intervention in Alzheimer’s disease. However, to date, very few studies have investigated the direct role of gamma-secretase in regulating cardiac mechanisms and in different cardiac diseases. This review will discuss the pathways directly and indirectly (through its substrates) modulated by gamma-secretase under normal and diseased states. By compiling proteins that are processed by gamma-secretase with desirable effects, it should be possible to modulate gamma-secretase activity and maximize the beneficial outcome. If this is possible, gamma-secretase could be a promising therapeutic intervention in the context of cardiac pathologies. This compilation from literautre is found in Table 1.

### 1.1. Gamma-Secretase Structure

Gamma-secretase is a multiprotein complex made up of four different protein subunits as shown in Figure 1. The principal part of gamma-secretase which imparts the complex with its catalytic activity is presenilin (PS). The other integral membrane proteins that contribute to the structure and function of gamma-secretase include nicastrin (NCT), anterior-pharynx defective 1 (Aph-1), and PS-enhancer 2 (PEN-2). These protein subunits are intricately involved in regulating each other at the intermolecular level, thereby indirectly regulating the entire gamma-secretase complex [5]. Interestingly, a fifth component, Basigin (CD147), is a non-essential regulator that can increase activity, highlighting a strong potential for effective gamma-secretase activators [6].
Figure 1Components of gamma-secretase processing a transmembrane substrate. (PDB ID: 5A63 from [7]).
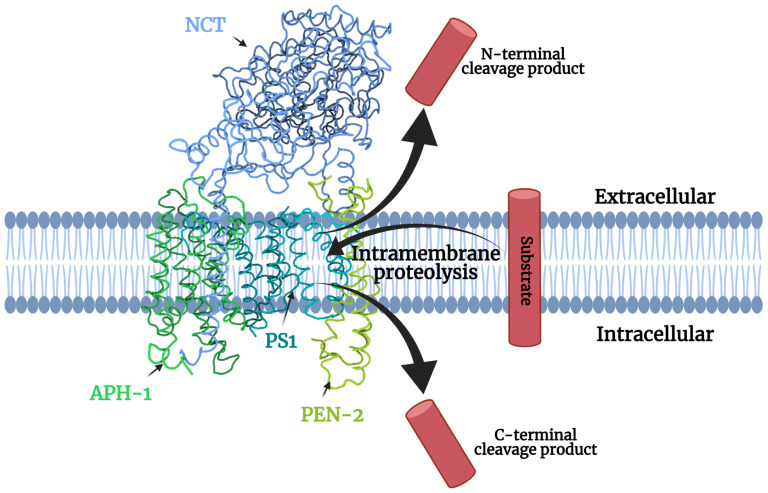

jpm-11-01294-t001_Table 1Table 1Summary of cardiac pathways that are subject to gamma-secretase direct regulation. The table summarizes the major findings from the literature that include gamma-secretase substrates, the cells they are expressed in and the diseases that substrate dysregulation has been linked to. The table also includes the potential beneficial effects of the substrates in the realm of cardiac disease as reported by the literature. We also indicate whether the experimental design suggests a preventive or curative approach for that particular disease. ROS—reactive oxygen species, LOF—loss of function, GOF—gain of function.Target PathwaysRole in the HeartTypes of Cell InvolvedDiseases Due to DysregulationPublicationsFindings Suggesting Beneficial Effects in Adult Cardiac DiseaseBeneficial Gamma-Secretase SubstratesNotchHeart developmentEndocardial and endothelial cellsCardiac fibrosis, heart failure, atherosclerosis, I/R injury, calcific aorta disease[8]LOF mutations lead to developmental defects[9]LOF mutations in endothelium lead to cardiac abnormalities[10]LOF mutations lead to vascular defects and embryonic lethalityCardiomyocytes[11]Overexpression and selective silencing has been correlated to developmental defects and lethalityAngiogenesis and vasculature maintenanceEndothelial cells[12]Endothelial-specific Jagged1 LOF mutants showed decrease in angiogenesisEndothelial-specific Jagged 1 GOF mutants showed increased angiogenesisCurative approach[13]Notch-1 mutants and Notch-1/ Notch-4 double mutants have defective angiogenic vascular development[14]LOF mutations in the NICD decrease angiogenesis. GOF mutations increase angiogenesis post I/R injuryCurative approachRegulation of survival and regeneration in I/R injuryCardiomyocytes[15]GOF ameliorated the increase apoptosis in cardiomyocytes seen in I/R injury and conferred cardioprotectionCurative approach[14]LOF and GOF mutations in the Notch intracellular domain showed decrease and increase in angiogenesis respectively in human umbilical cord cells post I/R injury as well as mice models post MICurative approachRegulation of cardiac fibrosisCardiac fibroblasts[16]Silencing of Notch-3 aggravates cardiac fibrosis in mice with MI as opposed to overexpressionPreventive approachCardiomyocytes[11]Cardiomyocyte specific upregulation of Notch-1 reduced fibrosis in post-MICurative approachErbB4Ventricular trabecular formationEndocardial cells cardiomyocytesMyocardial ischemia, systolic and diastolic heart failure[17,18,19]LOF causes embryonic lethalityRegulates ventricular wall developmentCardiomyocytes[17,18]LOF causes developmental disordersProliferation of cardiomyocytes[20]Inactivation of ErbB4 disrupted the normal proliferation of cardiomyocytes in postnatal miceRegulation of cardiac fibrosisCardiac fibroblasts[21,22,23]GGF2(recombinant NRG1) treatment post MI increased protection against fibrosisCurative approachAdaptation to changing heart demandsCardiomyocytes[20,24]Prior Inhibition leads to decreased adaptation to changing heart demands during MI and pregnancyPrevents systolic/diastolic heart failure[25]NRG1β attenuated the remodeling of ventricular wall and partially improved heart function in volume-overload HF miceCurative approach[26]Decreased expression correlates with terminal heart failure[27]LOF induces DCMKCNE1-4Maintenance of cardiac rhythmCardiomyocytesVentricular fibrillation, atrial fibrillation, long QT syndrome[28,29]KCNE Null and GOF Mutations associated with atrial-fibrillation[30]Missense Mutations associated with long QT syndromeNAVβ1-4Maintenance of cardiac rhythmCardiomyocytesVentricular fibrillation, atrial fibrillation, long QT syndrome[31]Loss of function mutations lead to sudden cardiac deathKlothoAttenuates ROSCardiomyocytesStroke, kidney disease-associated cardiovascular disease, cardiac hypertrophy[32]Overexpression improves cardiac function in aging, in endotoxemia, and reduces cardiomyocyte apoptosis in doxorubicin-induced injuryCurative approachMaintains ion homeostasis[33]Regulation of cardiac hypertrophy[34][35]Deficiency shows worsening of CV disease. Protective as shown to alleviate left ventricular hypertrophy. LOF studies show that Klotho null mice developed greater Left ventricular hypertrophy upon induction by indole-sulphatePreventive approach**Detrimental Gamma-Secretase Substrates**p75NTRRegulation of sympathetic innervationNeuronsReduced heart rate, Sudden cardiac death[36,37,38]LOF leads to reduced sympathetic innervation and synaptic transmission, downregulated basal heart rateDevelopment of microvascular injuryCardiac pericytesMicrovascular endothelial cellsMicrovascular injury and cardiomyopathy[39]Expression enables microvascular injury. LOF helps to rescues cardiomyopathyPreventive approach**Downstream Interactions**AMPK (activated by Notch)Regulation of cardiomyocyte growth and differentiationCardiomyocytesCardiac fibrosis, cardiac hypertrophy, heart failure[40,41]Silencing and mutations leads to increased cardiomyocyte hypertrophy, increasing expression attenuates cardiomyocyte hypertrophyPreventive approachRegulation of angiogenesisEndothelial cells[42]LOF inhibits angiogenesis and vascularizationManagement of ROS-induced damageCardiomyocytes Endothelial cells[43]LOF increased ROS-mediated fibrosis in response to isoproterenolPreventive approachRegulation of cardiac hypertrophy[44,45,46]Activation attenuates cardiac hypertrophy and improves survival and limits infarct sizeCurative approachRegulation of post-MI injury[47]

The sequential assembly of gamma-secretase in the order of Aph-1, NCT, PS, and PEN-2 is a prerequisite for the proper formation of the entire complex [48]. The initiation of the complex formation involves the interaction of the Aph-1 and NCT domain. The NCT sub-unit of gamma-secretase is characterized by post-translational modifications and is marked by heavy glycosylation [5]. It plays a vital role in the stabilization of gamma-secretase as well as helping in recognizing potential substrate molecules. The NCT complexes with Aph-1 and unitedly forms a scaffold-like unit [49]. Aph-1 stabilizes the scaffold and processing of substrates. This scaffold unit is responsible for the incorporation of PS. When PS is incorporated into the scaffolding unit, it constitutes the full-length inactive catalytic center. The recruitment of PEN-2 to the complex stimulates the endoproteolytic cleavage of the inactive, full-length, zymogen form of PS, abbreviated as PS-FL to the active form. The active form is characterized by the demarcation of the amino and carboxyl terminus domains of PS (PS-NTF/CTF), which remain joined together to form a heterodimer [5].

### 1.2. Gamma-Secretase Regulation

Active gamma-secretase constitutes only a small percentage of the total gamma-secretase population in cells, adding to the promising potential in the modulation of its activity [5]. The regulation of gamma-secretase can be carried out by an endoplasmic reticulum retrieval receptor (Rer1p) mediated biological pathway. Rer1p has been demonstrated to competitively inhibit Aph-1 binding with NCT [2], as it binds to the immature form of NCT. This binding causes NCT to be sent back to the endoplasmic reticulum, thereby preventing gamma-secretase complex formation. Rer1p has also been shown to interact with and bind to Pen-2 before its association with PS. A holoprotein named gamma-secretase activating protein (GSAP) has been linked to the modulation of gamma-secretase and shown as increasing its substrate affinity toward the amyloid precursor protein (APP) over other substrates, such as Notch [50].

The regulation of gamma-secretase can also be mediated by transcription factor hypoxia-inducible factor 1-alpha (Hif-1α), which plays an indispensable role in hypoxia-induced response [5]. Hif-1α shifts the gamma-secretase complex equilibrium from an inactive form to the activated form [51]. Besides these factors, studies have suggested that ERK1/2 is responsible for the downregulation of gamma-secretase. Inhibiting the ERK1/2 pathway, via methods ranging from MEK inhibitors to small RNA interference, has been shown to result in increased gamma-secretase activity. The activated ERK1/2 acts as both a transcriptional activator upon translocation to the nucleus and a cytosolic protein kinase to control cell signaling pathways. Studies have shown that there was no change in the protein expression levels of gamma-secretase complexes upon inhibition of extracellular signal-regulated protein kinase 1 and 2 (ERK1/2) due to unaltered gene expression [52]. However, upon the addition of purified active ERK1/2 to isolated gamma-secreted complexes, the gamma-secretase activity was found to be downregulated. Further studies showed that the phosphorylation of NCT by the cytosolic kinase activity of ERK1/2 was responsible for decreased expression of gamma-secretase [52]. This was due to phosphorylation-induced destabilization of NCT, leading to its proteasomal degradation and subsequently hampering gamma-secretase complex assembly [53].

### 1.3. Gamma-Secretase Function

Gamma-secretase exerts its action through the process of regulated intramembrane proteolysis, a type of membrane-to-nucleus signaling that involves cleavage of its substrates [54]. This process differs from the quintessential signaling cascades; in response to the activating stimulus, the receptors propagate the signal through intermediate messengers to the downstream targets in succession. In response to the stimulus activating gamma-secretase, the transmembrane substrate is subjected to regulated cleavage which gives to give rise to two biologically active signaling fragments on both sides of the cell membrane [5]. In some cases, the N-terminal cleavage product (extracellular) is mediated by other protease complexes, such as A disintegrin and metalloprotease 17 (ADAM/TACE) [55] and beta-secretase 1 (BACE1), while gamma-secretase cleaves the C-terminal (intracellular) product [56]. The extracellular product can signal to neighboring cells or participate in cell adhesion, whereas the intracellular product can initiate the downstream effects of that particular signaling pathway. These intracellular fragments can function as transcription activators or repressors, and regulators of cell cycle and cytoskeletal dynamics [57]. The lack of similarity in the structure of gamma-secretase substrates demonstrates the versatility of this enzyme, processing all of its substrates with the same mechanism [5].

## 2. Gamma-Secretase Animal Models

Although gamma-secretase is involved in the processing of many cardiac relevant substrates, few studies have looked into the exact role and potential of gamma-secretase in the context of cardiac disease. However, the findings from knockout models of gamma-secretase constituents can shed some light on the potential role of gamma-secretase in cardiac diseases. The ablation of PS, the catalytic center of the gamma-secretase complex, demonstrates the involvement of the gamma-secretase activity in cardiac development. PS1 knockout mice develop abnormalities in the development of the cardiac outflow tract. They show increased incidences of ventricular septal defects, as well as stenosis in the pulmonary artery [58,59]. A recent study indicated the role of PS1 as a direct regulator of the calcium pump in the heart [60]. On the other hand, PS2 knockout hearts were normally developed without signs of hypertrophy and fibrosis. They also showed abnormal calcium homeostasis and an increase in cardiac contractility, suggesting a role of PS2 in the excitation–contraction coupling [61].

Studies have also pointed out that homozygous knockout Aph-1 mice developed abnormalities in the heart and were embryonic lethal after E10.5. They also showed poor angiogenesis. The studies further established that the ablation of Aph-1 resulted in the gamma-secretase complex getting destabilized [62]. NCT knockout models have also been related to heart deformities and increased apoptosis in the heart and presented with a Notch deficiency-like phenotype [63]. The PEN-2 knockout mice were also shown to exhibit a Notch deficiency-like phenotype and heart abnormalities, implicating an absence of gamma-secretase activity and downstream Notch signaling [64]. The findings from ablation studies of various gamma-secretase subunits are summarized in Table 2.

## 3. Presenilin Mutations in Cardiomyopathy

Dilated cardiomyopathy (DCM) is a common form of cardiac disease. Studies have linked PS mutations to DCM and improper calcium signaling [68]. PS1 missense mutation, Asp333Gly, and PS2 missense mutation, Ser130Leu, are associated with DCM. PS2 Ser130Leu is also linked to peripartum cardiomyopathy [69]. The PS1 mutation causes severe impairment and presents as progressive heart disease, leading to complete cardiac failure. In some cases, this necessitates heart transplantation. The PS2 mutation presents as a milder form of DCM and cardiac failure. More interestingly, biopsy on patient hearts revealed no amyloid deposition, suggesting a pathologic mechanism independent of amyloid-β formation [68]. Therefore, gamma-secretase activity was affected by PS mutations [70]. It is possible that similar inhibition could contribute to cardiac dysfunction, further highlighting the importance of proper gamma-secretase function in cardiac health. In addition, the PS1 mutations -92delC, -21G>A, A953G/E318G, and PS2 mutation G185A/R62H are associated with idiopathic DCM [71]. With advancements in patient genetic screening, more mutations will likely be discovered in PS1, PS2, or other gamma-secretase subunits, which will allow researchers to understand how gamma-secretase dysfunction contributes to cardiac diseases.

## 4. Substrates of Gamma-Secretase—Implications for Cardiac Disease

Gamma-secretase cleaves over 100 known substrates, which are highly varied in function [49,56,72]. This paper focuses on gamma-secretase substrates and downstream pathways potentially associated with cardiac diseases. We have included the Notch pathway, ErBb4, KCNE proteins, Klotho, channel proteins, and P75${}^{NTR}$ (see Table 1). From lipid metabolism, cell proliferation, apoptosis, voltage-gated channels, angiogenesis and energy regulation, gamma-secretase substrates play an instrumental role in overall cellular homeostasis.

### 4.1. Notch Pathway

The Notch pathway (Notch) is a crucial intercellular crosstalk pathway during organism development, which is mediated by the interaction between Notch receptors (Notch1–4) and its ligands (Jagged1, and -2 and Delta1, -3, and -4). Notch receptors and Notch ligands (Delta1 and Jagged2) are gamma-secretase substrates [73]. Delta1 and Jagged2 both activate Notch [74]. The exact role of Delta1 and Jagged2 intracellular products after gamma-secretase processing is unclear, but it is hypothesized that they provide additional signaling to regulate Notch [73].

At first, Notch receptors are expressed in the Golgi in the inactive form. Then, the furin protease carries out the initial processing of Notch, mediating the S1 cleavage of the inactive Notch receptor in the Golgi. Next is glycosylation by the Fringe family of glycosyltransferases. Notch receptors translocate and integrate into the cell membrane in a heterodimeric confirmation. The Notch extracellular domain (NECD) projects out from the cell membrane and interacts with the Notch ligands expressed on the surfaces of the neighboring cells [75]. Upon interaction with the Notch ligand, the receptor is acted upon by ADAM10, which causes ectodomain shedding and the subsequent release of NECD by an S2 cleavage event. This is shown in Figure 2.

On the other hand, in the cells that are expressing the Notch ligand, an E3 ubiquitin ligase enzyme plays a crucial role in the ubiquitination mediated uptake of the Notch ligand [76]. The receptor, after ectodomain shedding, presents as the substrate for gamma-secretase. The gamma-secretase catalytic center carries out the S3 cleavage of the Notch receptor. This cleavage event causes the dissociation of the membrane-spanning domain of gamma-secretase from the Notch intracellular domain (NICD). S3 cleavage is followed by the nuclear translocation of the NICD, where it functions as a transcriptional activator. The NICD forms an activation complex with another activator protein, Mastermind (MAML). This complex serves to override the repression of transcriptional activity by CSL/RBPJ/Su(H), leading to the activation of downstream genes such as Hes, Hey, p21, c-myc, and more [76].

Notch plays an important role in tissue homeostasis as well as developmental processes, such as cell differentiation, cell migration, and the regulation of apoptosis [77]. Notch has been shown to participate in all stages of embryonic heart development [78] including in angiogenesis and the maintenance of vasculature [79]. Studies have established a role of Notch in endocardial cells for the formation of the heart valve by endothelial to mesenchymal transition [80]. Abnormalities in Notch are linked to congenital heart diseases. Notch, though largely inactive in the adult heart, becomes upregulated in response to cardiac pathologies [77].

Data suggest that Notch protects against myocardial damage resulting from ischemia/ reperfusion (I/R) injury. Notch has also been implicated in angiogenesis after I/R injury [14]. Vascular endothelial cells have been shown to express a number of Notch signaling players including Notch1, Notch4, Jagged1, and Delta-like 4 (Dll4). Notch signaling has distinct roles during different developmental stages of the heart. After vascular injury, Notch1 has been shown to be activated and to upregulate vascular smooth muscle cell growth as well as neointimal formation in the injured myocardium. Vascular endothelial cells upregulate the secretion of vascular endothelial growth factor (VEGF) to induce angiogenesis after I/R injury [81]. Studies in mouse embryos have shown that VEGF expression increased the expression of Dll4 resulting in subsequent neoangiogenesis as well as the preservation of vascular integrity [82]. In addition, Dll4–Notch1 is responsible for angiogenesis in the coronary plexus of the inner myocardium in mouse embryos [83]. However, Jagged1 plays an important role in limiting excess neovascularization by inhibiting the activity of angiogenesis promoting Dll4 [12]. Overall, Notch plays a role in balancing neoangiogenesis and neovascularization via Dll4 and Jagged1 [84].

Notch1-deficient mice show more severe myocardial infarction (MI) and have unfavorable cardiac parameters as compared to wild-type mice [85]. This indicates that Notch plays an important role in preventing the decline of heart function in myocardial ischemic hearts. Ischemic pre- conditioning were shown to be associated with various signaling pathways that confer cardioprotective activity after I/R injury [86]. It was observed that Notch signaling activation coincides with both ischemic pre- postconditioning and contributes to increased viability of cells. Notch1 signaling was shown to reduce oxidative damage and restore heart function during I/R by upregulating Hif-1a expression via the Hes1/STAT3 pathway [87]. NICD knockdown-mediated inactivation of Notch was shown to nullify ischemic pre- and postconditioning-induced protective activity [15]. Hence, there is potential in using Notch signaling as a possible therapeutic mimic for conferring cardioprotection in I/R injury. Therefore, gamma-secretase could be modulated to affect Notch activity as a potential target against ischemic injury.

Notch signaling has been correlated with a reduction in cardiomyocyte hypertrophy, increased division of progenitor cells in the heart, and maintenance of cardiac tissue differentiation [15]. Notch is involved in the anti-fibrotic mechanisms mostly in the cardiac fibroblasts. Notch inhibition increases cardiomyocyte apoptosis and cardiac hypertrophy, which characterizes and accelerates the progression of cardiac fibrosis [88,89]. After injury to the heart, such as MI, loss in the number of cardiomyocytes initiates an immune response, which results in the increased expression of neutrophils that act in the damaged area of the heart. This, in turn, upregulates the expression of macrophages that promotes the secretion of fibrosis-promoting cytokines, such as the transforming growth factor-β (TGF-β). Increased levels of TGF-β results in the activation of the TGF-β/Smad3 pathway that increases the conversion of cardiac fibroblasts to myofibroblasts, a hallmark of cardiac fibrosis. The myofibroblasts contribute to increased secretion and deposition of the extracellular matrix as well as collagen type I that disrupts the myocardial architecture and establishes cardiac fibrosis [89]. The process of fibrosis initiates as a compensatory process to cardiac damage. However, in the long term, fibrosis will contribute to cardiac dysfunction [90]. Notch acts antagonistically on cardiac fibrosis. In neonatal hearts, the conversion of fibroblast to myofibroblast was accompanied by the decrease in Notch1, -3, and -4 expressions [91], although the blocking of Notch signaling has been shown to coincide with myofibroblast conversion. The TGF-β/Smad3 pathway is the primary pathway responsible for promoting the activation of cells and subsequent fibrogenesis. Notch helps prevent fibrosis by inhibiting TGF-β and Smad3 pathways via Notch3 [16].

With respect to cardiac fibrosis, the Notch pathway could be manipulated and used as a curative. Overexpressing Notch by upregulating its activation by manipulation of upstream gamma-secretase activity could help inhibit the progression and establishment of cardiac fibrosis by Notch-mediated inhibition of the TGF-β/Smad pathway.

### 4.2. ErbB4

The epidermal growth factor receptor tyrosine kinase falls within the category of transmembrane receptor tyrosine kinase. Its member ErbB4 is a gamma-secretase substrate. The recognition of receptor tyrosine kinase substrates, such as ErbB4, is postulated based on the stability and strength of the interaction between the transmembrane domain of the substrate and the gamma-secretase complex [92]. It has also been linked to the length of the ectodomain post shedding [93] and localization of the substrate with respect to gamma-secretase activity [94]. However, all of the details of the various structural elements partaking to help gamma-secretase recognize receptor tyrosine kinases, such as ErbB4, need further investigation [95]. Similar to Notch, ErbB4 activation depends on intramembrane proteolysis by gamma-secretase. Upon binding the growth factor ligand, Neuregulin (NRG-1) ADAM17/TACE mediates the cleavage of ErbB4. This cleavage event fragments the ErbB4 into two different parts. The first is the 120 kDa ectodomain which is released into the extracellular space. The other portion, 80 kDa in size, consists of a combination of residual ectodomain residues, the complete transmembrane subunit, and the cytoplasmic domain [96]. Analogous to the Notch receptor, the ErbB4 undergoes a second round of proteolysis. The gamma-secretase mediates this at the transmembrane domain, and the cytosolic intracellular domain (ICD) is released [97]. The nuclear translocation of this domain and its subsequent binding to transcription factors alter the transcription of the downstream target genes. Many processes are partly dependent on the nuclear ErbB4 ICD. It has been postulated that the ErbB4 ICD increases the proliferation and growth of mesenchymal cells during valvulogenesis in the mouse heart [98].

Neuregulin (NRG1) belongs to the family of epidermal growth factors and is very relevant in the context of the heart. As discussed previously, NRG1 interacts with members of the epidermal growth factor receptor family, including ErbB2, ErbB3, and ErbB4. It plays a significant role in the development of the cardiac system. In the prenatal phase, NRG-1 expression is localized to the endocardium, whereas ErbB2/4 are localized to the ventricular myocytes [99]. NRG1 is associated with Erb2 and Erb4, which communicate with bone morphogenetic protein-10 (BMP-10) to modulate the process of ventricular trabecular formation [15]. Studies have shown that Notch signaling plays a role in the expression of BMP-10 as well as ErbB2/4 and concomitant positive modulation of ventricular trabecular formation [15]. Studies utilizing knockout models of NRG1, ErbB2, and ErbB4 showed abnormal ventricular wall development at E9.5, which led to fetal death. In studies using cultured neonatal rat cardiomyocytes, NRG1 addition was shown to induce DNA synthesis [100]. The NRG1-ErbB2/4 complex was also shown to upregulate the differentiation and speciation of cardiac progenitor cells into cardiomyocytes [100]. Studies using NRG1 mutated zebrafish models showed that the defects led to a reduction in the number of cardiomyocytes, a decrease in the density of the intracardiac myocardium, as well as improper regulation of the heartbeat [101]. It was established that NRG1 plays a vital role in the formation of the cardiac nerve plexus, thereby exerting an effect on cardiac development [101]. The NRG1-ErbB2/4 also mediates the conversion of fetal cardiac cells into pacemaker cells [99].

In adult hearts, NRG1 signaling plays a different role than it does in prenatal cells. NRG1 mediates ErbB2 and ErbB4 to mediate downstream signaling pathways, such as MAP kinase, endothelial nitric oxide synthase, as well as protein kinase B (AKT) pathways [22]. NRG1-ErbB4 is cardioprotective against I/R injury and crosstalks with the AKT pathway to impart further cardioprotection [99]. It was established that the NRG1-ErbB2/4 pathway is indispensable for the adaptation of the heart to changing cardiac demand [102]. NRG1 and ErbB2/4 are located in the vascular endothelial cells and participate in the angiogenesis response. Deletion of NRG1 in mouse models led to deformities in the endocardial cushion, showing the importance of ErbB signaling in the development of vascular endothelium [103]. They function in the prevention of apoptosis of progenitor cells in the endothelium as well as stimulating pro-survival signaling pathways [21]. NRG1 was shown to upregulate VEGF expression and take part in arteriogenesis. Lastly, the NRG1 system partakes in decreasing active myofibroblast formation and attenuating collagen synthesis, thereby preventing fibrosis [21]. Besides this, NRG1-ErbB2/4 was linked to the prevention of systolic and diastolic heart failure [25]. Due to all of these beneficial effects of NRG1 and ErbB2/4, this system can be taken advantage of in designing therapeutics as well as promoting myocardial regeneration [20]. In this regard, it will be intriguing to examine whether modulating gamma-secretase to regulate NRG1-ErbB2/4 and Notch pathways can offer better cardioprotective effects.

### 4.3. Voltage-Gated Channels

Voltage-gated sodium (Nav) and potassium (Kv) channels are the main channels in maintaining cardiomyocyte action potential, and therefore are essential for cardiac rhythm control [104]. KCNE1–4, voltage-gated potassium channel subunits with a wide variety of functions, are substrates of gamma-secretase. Gamma-secretase regulates endogenous KCNE1 and KCNE2 after alpha-secretase or BACE1 cleavage, releasing an ICD for signaling. This activity is thought to regulate Kv currents, including the delayed-rectifier K${}^{+}$ current [105]. Thus, it plays a vital role in cardiomyocyte repolarization, where dysfunction leads to numerous pathophysiological states, including ventricular fibrillation, atrial fibrillation, and long QT syndrome [106,107]. Nav-β 1–4 are subunits processed by gamma-secretase that modulate gating, kinetics, and localization of voltage-gated sodium channels [108]. The intracellular product from gamma-secretase cleavage of Nav-β2 regulates the surface levels of sodium channels and was shown to promote an increase in Nav1.1 mRNA and protein [52,109]. It is possible that KCNE processing might modulate Kv currents in an analogous way [105]. These examples suggest that cleaved substrates interacting with other subunits and voltage-gated channels are essential crosstalk for maintaining a proper cardiac rhythm. Ample amounts of active gamma-secretase may be vital for voltage-gated channel regulation. When these details are elucidated, gamma-secretase could be employed to help regulate voltage-gated channels in arrhythmia settings.

### 4.4. Klotho

Klotho is generally known as an anti-aging protein and functions as a humoral factor with pleiotropic activities [110]. While many physiological functions of klotho are associated with its membrane-bound form as a type 1 membrane protein, its extracellular region can be shed and processed by gamma-secretase [111]. In general, secreted klotho exerts anti-aging and cardioprotective effects by attenuating reactive oxygen species (ROS) and maintaining ion homeostasis [110]. It was found to reduce stress-induced cardiac hypertrophy in cardiomyocytes [112] and inhibits angiotensin-2–induced cardiac hypertrophy, fibrosis, and dysfunction in mouse models [113]. Processing by gamma-secretase yields a 5 kDa product from Klotho stub [111]. The exact function of the cleavage and klotho products from gamma-secretase is an ongoing section of research. However, because elevated amounts of klotho are generally cardioprotective [114], modulating gamma-secretase to increase the cleaved form of klotho could be a feasible therapeutic strategy in the future.

### 4.5. p75^NTR^

p75${}^{NTR}$ is a type 1 transmembrane protein that binds neurotrophins. It is partially responsible for ensuring the correct density of neurons by modulating cell survival pathways [115]. p75${}^{NTR}$ shedding and gamma-secretase cleavage is required for various aspects of the p75${}^{NTR}$ signaling cascade, and evidence shows that inhibiting gamma-secretase inhibits certain p75${}^{NTR}$ functions [116]. Mice with a lack of function for p75${}^{NTR}$ exhibit peri-infarct sympathetic denervation after cardiac I/R [38]. p75${}^{NTR}$ was investigated in cardiac development, potentially playing a role by analyzing sympathetic and parasympathetic nervous tissue [36]. This function is essential for contraction, as cardiac sympathetic neurons are stimulated through the release of norepinephrine [37]. Interestingly, studies show that deficiency in p75${}^{NTR}$ limits the infarct size after I/R injury. p75${}^{NTR}$ expression in pericytes is remarkably important for the development of microvascular injury [39]. Therefore, the exact role of p75${}^{NTR}$ in the context of cardiac pathologies is unclear. Still, temporal and specific inhibition of gamma-secretase to change p75${}^{NTR}$ activity may be beneficial to treat MI.

## 5. Downstream Interaction: Notch, AKT, AMPK

A large number of receptor tyrosine kinases also serve as substrates to gamma-secretase [117]. Receptor tyrosine kinases such as VEGF and tyrosine kinase with immunoglobulin-like and EGF-like domains 1 (TIE1) that are gamma-secretase substrates were seen to be the upstream activators of the AKT pathway [118,119]. Studies have implicated the role of constitutively active AKT and mTOR in adaptive or physiological cardiac hypertrophy that happens in response to physiological conditioning, such as exercise and normal growth. Disruption of this pathway is linked to pathological cardiac hypertrophy [120]. AKT contributes to cardiomyocyte size regulation in association with mTOR-dependent growth-enhancing pathways [121] as well as inhibition of the WNT pathway signaling component, GSK3β [122]. The AKT pathway also increases the phosphorylation and activation of SERCA2a, as well as inhibiting PLN, regulating heart contractility [123]. However, long-term activation of the AKT pathway, specifically AKT1, leads to cardiac failure characterized by decreased angiogenesis, increased pathological hypertrophy, and pathological left ventricular remodeling [124]. Thus, the timing of AKT expression and activation is pivotal for cardiac health.

Studies performed on various species have established Notch1 to be the downstream target of the VEGF signaling [125]. VEGF was shown to modulate the upregulation of the AKT pathway in endothelial cells, leading to increased angiogenesis and vasorelaxation via increased Notch1 expression. Notch1 was shown to mediate cardioprotection via the AKT-mTOR-Stat 3-Notch1 cascade [126]. Therefore, gamma-secretase could modulate both Notch and AKT pathways and possibly be a therapeutic intervention point to prevent prolonged expression of AKT and consequent heart pathologies.

The AMP-activated protein kinase (AMPK) is a key metabolic regulator that functions to control the cellular energy status of the heart [127]. Studies have shown that the AMPK pathway interacts with Notch. Further investigations using immunoprecipitation experiments suggested that liver kinase B1 (LKB1), the upstream activator of AMPK, be a point of crosstalk with the Notch pathway. Downregulation of Notch leads to reduction in AMPK activity; therefore, it was inferred that Notch1 could mediate the AMPK pathway by mediating its upstream activator kinase LKB1 [128]. When the cell is deficient in energy, AMPK upregulates the energy-producing processes while selectively downregulating energy-depleting pathways. The AMPK is termed the “low fuel warning system” for its regulation of energy levels in cells [129]. Cellular oxidative stress also activates AMPK. Energy regulation is especially crucial in the heart, which uses around 6 kg of ATP per day with a limited presence of energy reserves [130]. In the context of heart diseases, there were some studies that showed the cardioprotective role of the AMPK pathway. AMPK is associated with the regulation of cardiomyocyte growth and division. It also positively regulates the process of angiogenesis [42] and the eNOS pathway [131]. AMPK was shown to cross-talk with AKT to regulate angiogenesis in hypoxic conditions [42]. AMPK has a cardioprotective function against ROS-induced damage in cells [75]. AMPK was seen to prevent palmitate-induced apoptosis of endothelial cells by downregulating the production of ROS [132]. AMPK activation could reduce cardiac hypertrophy by reducing protein O-linked N-acetylglucosaminylation [44]. AMPK also functions in limiting myocardial fibrosis (see Table 1) and pathological remodeling of the left ventricle by antagonizing pro-fibrotic signals. AMPK helps minimize post-MI injury by aiding in the production of a mature collagen scar [47]. In summary, upregulating AMPK, with its protective roles and mechanisms, should be beneficial for treating cardiac diseases. This Notch–AMPK crosstalk warrants further investigation to examine whether we can garner the cardioprotective activity of AMPK by modulating gamma-secretase activity.

### Downstream Complexity

Figure 3 is an example of how complicated protein–protein interaction networks can become with only a small number of proteins (nodes). After inputting substrates of interest into STRING and including just 16 other interacting partners, the output has 166 interactions (edges). These other interacting partners highlight some of the downstream effects of gamma-secretase and its substrates. In this small subsection from Figure 3 is the regulation of energy homeostasis, heart development, cell determination, voltage-gated channel regulation, proteostasis, and fibrosis-related pathways. The growth of edge number as nodes increase in protein–protein interaction networks is astronomical, and including all gamma-secretase substrates, creates an extremely complicated network. Advancements in RNAseq technology and proteomics may make this type of analysis possible, but these bioinformatic experiments are yet to come for many gamma-secretase modulators.

## 6. Difficulty in Drug Development

In general, drugs that target gamma-secretase can be classified into six categories: gamma-secretase inhibitors (GSIs), non-steroidal anti-inflammatory agents gamma-secretase modulators (NSAID-GSMs), 2nd generation acidic gamma secretase modulators, non-acidic modulators, triterpenoid compounds, and inverse gamma secretase modulators (iGSMs) [137]. Some common GSIs and GSMs in development are listed in Figure 3. GSIs inhibit gamma-secretase activity, GSMs act as processivity enhancers, and iGSMs as inhibitors of processivity [138,139]. Depending on the modulator, this change in processing can be due to binding to gamma-secretase subunits or substrates [137]. For example, allosteric GSMs can specify Aβ42 over Aβ40 [140]. There are even notch-sparing gamma-secretase inhibitors which were studied in the context of Alzheimer’s disease, inhibiting APP preferentially relative to Notch1 [72]. These compounds are at the forefront of Alzheimer’s disease and cancer research, but after decades, gamma-secretase modulators are still far from being understood in the full biological context.

Despite a massive effort from the research community, challenges in the timing, selectivity, and downstream effects have led to all clinical trials of GSIs ending prematurely [72]. On top of that, there is a lack in translation from in vitro to in vivo pharmacology for GSMs [72]. Currently, the clinical efficacy is at question with considerable side effects [141]. Currently, inhibitors DAPT, Avagacestat, MK-0752, Nirogascestat, RO4929097, Begacestat, Semagacestat, LY-411,575, and LY-900009 are in preclinical or a stage of clinical trials [142]. The farthest along in human testing is the GSM CHF5074, which has completed a phase II trial [143].

Another challenge for applying GSMs to cardiac diseases also comes from the focus of treatment; with the focus on Alzheimer’s disease and cancer. This is apparent in Table 2, where only limited investigated drugs have been studied with cardiac responses. For example, the GSM MH84 has favorable effects for Alzheimer’s and increases mitochondrial respiration, but has not been studied in the context of the heart [136]. Positive protein GSMs, CD147 and GSAP, are associated with cardiovascular inflammation and valvular calcification, respectively [65,66], which suggest that the non-selective promotion of gamma-secretase activity may not be favorable. On the other hand, GSI DAPT inhibits differentiation, which is also not a desirable effect [67]. Therefore, positive GSMs and GSIs alike can have undesirable effects on the heart. It is possible that gamma-secretase, such as Notch, requires goldilocks activity [144]. That is, too much activity or too little activity can be unfavorable. This is where a “precision medicine” approach, modulating gamma-secretase activity for certain substrates, could be beneficial [145].

One of the main issues with GSMs and iGSMs for Alzheimer’s treatment is potency and brain penetrance with the restriction of the blood–brain barrier [137], which is not applicable to the heart. Thus, it could be a promising area to test the effect of these pre-established GSMs in treating cardiac diseases. Given the scarcity of data on cardiac gamma-secretase modulation, the findings from these studies can potentially shed more light on the feasibility of targeting gamma-secretase in various forms of cardiac diseases.

## 7. Conclusions

Overall, gamma-secretase enables a plethora of cell signaling through intramembrane proteolysis. When gamma-secretase function is disrupted, by mutations or knockouts, it is generally detrimental to heart function. This makes sense intuitively because of the role of gamma-secretase substrates in cardiac development and regulation. Still, contrasting studies using inducible PS knockout models suggested that deleting the gamma-secretase complex helped improve angiotensin-II mediated left ventricular hypertrophy [146]. Despite this, gamma-secretase substrates, such as Notch, neuregulin, and klotho, are generally found to be beneficial for cardiac health (see Table 1). The only substrate we explored that did not have cardioprotective effects in abundance was p75${}^{NTR}$, where lower concentrations can improve myocardial function in mouse models [39]. However, the total effect of the p75${}^{NTR}$ presence enabling injury is unclear, so it is hard to conclude whether this would be beneficial.

It is not currently clear whether gamma-secretase modulators or inhibition will have a beneficial outcome due to the regulation of substrates. However, it is essential to note that while modulating gamma-secretase may be beneficial for certain cardiac pathologies, it may complicate other disease states, such as Alzheimer’s disease. The accumulation of β-amyloid, where the formation is catalyzed by gamma-secretase, is a hallmark in the development of Alzheimer’s disease [147]. Therefore gamma-secretase inhibitors have been studied feverishly to treat Alzheimer’s disease. Unfortunately, like many proteins, total inhibition leads to serious side effects. More recent is the “precision medicine” approach, where the development of drugs focuses on modulating activity away from neurotoxic products while enabling maintenance of homeostasis [145]. The possibilities with this approach are bountiful and intriguing. The development of the precision modulators could be tailored to multiple pre-selected gamma-secretase pathways to target specific cardiac diseases. It is foreseeable that this approach will maximize the beneficial effects and limit undesirable side effects.

Additionally, cellular signaling pathways are incredibly vast, and many of these substrates might be caught up in some beneficial or harmful pathways. Out of more than 100 substrates known to be processed by gamma-secretase, some substrates are not usually expressed in the heart (e.g., ApoER2) [148], while others have functions that do not seem related to heart function at all (e.g., TYRP1, melanin synthesis) [149]. Thus, we are still lacking a complete picture of the gamma-secretase regulatome. We expect that with the advancement in proteomic and transcriptomic analyses, researchers will be able to dissect protein–protein interaction networks and identify the optimal “switch” on gamma-secretase, which will allow the correction of multiple disease mechanisms. Based on the current findings, proper gamma-secretase function appears to be protective in various disease conditions, which warrants attention to further examine its potential as a therapeutic remedy in cardiac diseases.

## Figures and Tables

**Figure 2 jpm-11-01294-f002:**
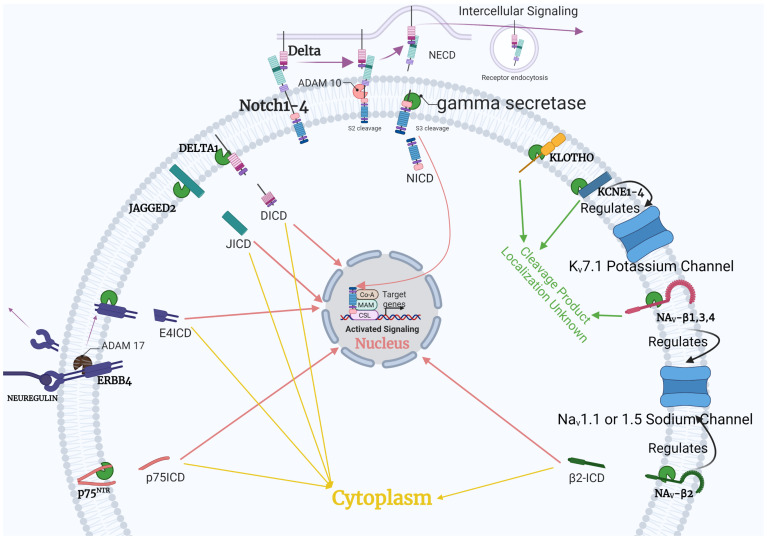
Processing of the cardiac-relevant substrates by gamma-secretase activity. The cell-type specific findings can be referred to Table 2. The in-depth explanation of Notch signaling serves as an example for the generalized gamma-secretase function. The red and yellow arrows show where intracellular domains (ICDs) from cleaved products localize in the cell. Products with unknown localization are labeled with green arrows.

**Figure 3 jpm-11-01294-f003:**
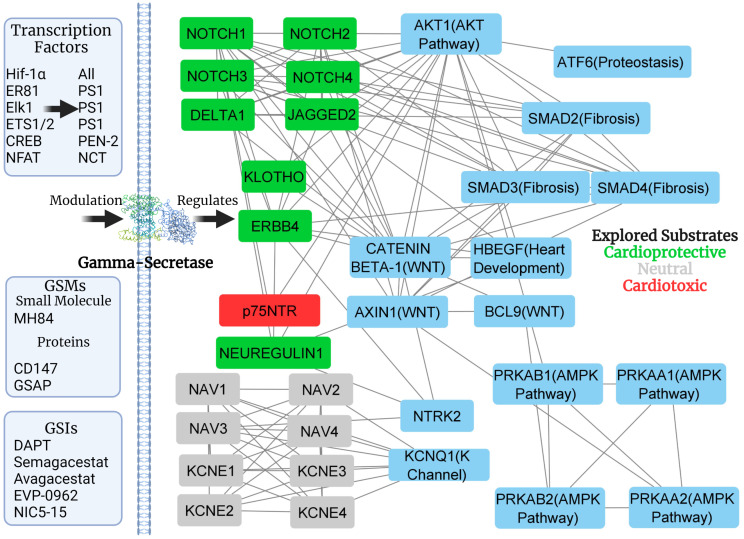
Examples of transcription factors [51,133,134] (and corresponding targets), gamma-secretase modulators (GSMs) [6,50,135,136], and inhibitors (GSIs) with the potential of changing gamma-secretase activity; affecting the subsequent inter-connected signaling pathways. The protein–protein interaction network includes the substrates of interest and 16 other interacting partners to create a network with a path between every node.

**Table 2 jpm-11-01294-t002:** Gamma-secretase alterations and corresponding cardiac phenotypes. The table summarizes gamma-secretase knockout models, some gamma-secretase modulators (GSMs), and inhibitors (GSIs) that are reported in the literature. Information on how the GSMs and GSIs impact cardiac diseases warrants further studies.

Potential Gamma-Secretase Alterations	Cardiac Disorders Observed	Publication
**GAMMA-SECRETASE KNOCKOUT MODELS**		
Presenilin 1	Cardiac outflow tract development defect, ventricular septal defects, Stenosis of the pulmonary artery	[59] [58]
Presenilin 2	Increased cardiac contractility, abnormal calcium homeostasis	[61]
Nicastrin	Heart development abnormalities and increased apoptosis in abnormally developed regions Embryonic lethal	[63]
Aph-1	Reduced angiogenesis Embryonic lethal	[62]
PEN-2	Notch deficiency-like phenotype, poor embryonic development of heart Embryos have a large pericardial sac	[64]
**GAMMA-SECRETASE MODULATORS (PROTEIN)**		
CD147	Promotes cardiovascular inflammation, myocardial remodeling, and myocardial I/R injury	[65]
GSAP	Valvular calcification	[66]
**GAMMA-SECRETASE MODULATORS (SMALL MOLECULES)**		
MH84	No cardiac adverse effects have been reported in these agents yet	Not reported yet in literature
**GAMMA-SECRETASE INHIBITORS**		
DAPT	Prevents differentiation in development	[67]
Semagecestat	No cardiac adverse effects have been reported in these agents yet	Not reported yet in literature
Avagacestat		
EVP-0962		
NIC5-15		

## Data Availability

Not applicable.

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
