# Peer review of "The Potential of Gamma Secretase as a Therapeutic Target for Cardiac Diseases"

_jpm, 2021, doi:10.3390/jpm11121294_

Round 1
Reviewer 1 Report
This article from Sen et al., reviewed previous literature on the topic that gamma secretase is a therapeutic target for heart disease. Although the direct role of gamma secretase in heart disease remains elusive, it has been shown to process a number of substrates that are relevant to the progression of heart disease. The authors provided an adequate introduction to the structure, activity regulation and function of gamma secretase. They then summarized the findings on substrates for gamma secretase that are implicated (mostly beneficial) to heart disease. The relevance of gamma secretase in cardiac pathology is further supported by the presence of presenilin mutations in cardiomyopathy. Overall, this is a well-written review linking the biochemistry of gamma secretase and its substrates with cardiac pathology implications. Below are my specific comments:
Comments
- What is known about gamma-secretase on the regulation cardiac physiology/pathology? The authors suggested that there are very few studies. Are there KO models available for the proteins that make of beta-secretase? If so, are there any cardiac phenotypes in these KO mice?
- In lines 79-81, what are the two ‘it’ referring to need to be specified.
- In line 156, ‘increased expression of the players of Notch has been correlated …’. Which are the players in Notch signaling that were referring to here?
- In line 206, what is (Lee, 2001) referring to?
- It will be helpful to add a table that categorizes the protective and detrimental substrates.
Author Response
This article from Sen et al., reviewed previous literature on the topic that gamma secretase is a therapeutic target for heart disease. Although the direct role of gamma secretase in heart disease remains elusive, it has been shown to process a number of substrates that are relevant to the progression of heart disease. The authors provided an adequate introduction to the structure, activity regulation and function of gamma secretase. They then summarized the findings on substrates for gamma secretase that are implicated (mostly beneficial) to heart disease. The relevance of gamma secretase in cardiac pathology is further supported by the presence of presenilin mutations in cardiomyopathy. Overall, this is a well-written review linking the biochemistry of gamma secretase and its substrates with cardiac pathology implications. Below are my specific comments:
Response:
We sincerely thank the reviewer for the encouragement and have made changes to the manuscript according to your comments, detailed below.
What is known about gamma-secretase on the regulation cardiac physiology/pathology? The authors suggested that there are very few studies. Are there KO models available for the proteins that make of beta-secretase? If so, are there any cardiac phenotypes in these KO mice?
Response:
Thanks to the reviewer for your valuable comment. We greatly appreciate you pointing this out. There are knockout models for the gamma-secretase complex. Ablation of gamma-secretase components, presenilin 1/2, nicastrin, PEN2 as well as Aph-1 have been linked to cardiac defects during development. For instance, presenilin 1 knockout mouse models were shown to develop improper cardiac outflow tract, whereas presenilin 2 models were shown to exhibit an enhancement in cardiac contractility. In many cases, the phenotypes of Aph-1, nicastrin, and PEN2 knockout models were very similar to Notch-deficiency phenotypes and involved embryonic cardiac developmental effects. Studies have connected these phenotypes to a lack of processing of Notch by gamma-secretase, due to the knockout of the constituents of the gamma-secretase complex and the resultant inactivation of it. The details of the knockout models and the corresponding phenotypes in the cardiac system have been added to the manuscript under section 2 titled ‘Gamma-secretase and Animal Models’ as well as in Table 2.
In lines 79-81, what are the two ‘it’ referring to need to be specified.
Response:
We thank the reviewer for pointing out the need for clarity. We have made changes to ensure Hif-α is the clear subject of these sentences.
In line 156, ‘increased expression of the players of Notch has been correlated …’. Which are the players in Notch signaling that were referring to here?
Response:
Thank you for pointing out the need for clarification. We have incorporated the names of the players and made necessary changes to the manuscript (Lines 206-219). Notch1 has been shown to be activated and upregulated in response to cardiac injury, as in the case of myocardial infarction and I/R injury. Notch 1 plays a role in augmenting vascular smooth muscle cell growth and neointimal formation in the injured myocardium. Vascular endothelial cells partake in increased secretion of vascular endothelial growth factor (VEGF) that in turn induces angiogenesis after I/R injury. Increased VEGF expression has been shown to augment the expression of Delta-like 4 (Dll4). Dll4 works to enhance neo-angiogenesis and preservation of vascular integrity. Notch player Jagged1 functions to limit excess neo-vascularization by inhibiting angiogenesis promoting activity of Dll4. Hence, Notch plays a role in balancing neo-angiogenesis and neovascularization via Dll4 and Jagged1.
In line 206, what is (Lee, 2001) referring to?
Response:
Thanks to the reviewer for noticing this. We have replaced this typo with the correct citation and reference. This is now in line 278.
It will be helpful to add a table that categorizes the protective and detrimental substrates.
Response:
We thank the reviewer for the valuable input. We have organized the beneficial and detrimental substrates in the manuscript in Table 1. In the table, we have collated all the important findings from the literature that has been mentioned in the paper. The table entails information about the gamma-secretase substrates, their roles in the heart, the corresponding cell types as well as cardiac diseases caused due to dysregulation of these substrates. We also included the overall protective and detrimental effects of these substrates in the field of cardiac diseases. The table also includes downstream-interacting partners of gamma-secretase substrates like AMPK.
Reviewer 2 Report
To:
Editorial Board
Journal of Personalized Medicine
Title: “GAMMA SECRETASE AS A THERAPEUTIC TARGET FOR CARDIAC DISEASE”
Dear Editor,
I read this paper and I think that:
- The authors should include a table gathering the main findings from literature. Please provide.
- The authors should also include a table resembling the possible cardiac diseases involved in gamma secretase alterations.
Author Response
The authors should include a table gathering the main findings from literature. Please provide.
Response:
We thank the reviewer for the valuable comment. We have summarized all the relevant findings from the literature in Table 1. In the table, we have collated all the important findings from the literature that has been mentioned in the paper. The table entails information about the gamma-secretase substrates, their role in the heart, the cell types they function in as well as cardiac diseases caused due to dysregulation of these substrates. We also included their overall beneficial and detrimental effects as reported in literature. The table also includes downstream-interacting partners of gamma-secretase substrates like AMPK.
The authors should also include a table resembling the possible cardiac diseases involved in gamma secretase alterations.
Response:
We greatly appreciate your feedback. We have collated the asked information on possible cardiac diseases involved in gamma-secretase alterations under Table 2. This table summarizes the gamma-secretase alterations (knockout models, modulators, agonists, and inhibitors) that have been reported in the literature and the consequent cardiac phenotypes found. For instance, presenilin 1 knockout mouse models were shown to develop improper cardiac outflow tract, whereas presenilin 2 models were shown to exhibit an enhancement in cardiac contractility. In many cases, the phenotypes of Aph-1, nicastrin, and PEN2 knockout models were very similar to Notch-deficiency phenotypes and involved embryonic cardiac developmental effects. Studies have connected these phenotypes to a lack of processing of Notch by gamma-secretase, due to the knockout of the constituents of the gamma-secretase complex and the resultant inactivation of it.
Reviewer 3 Report
Reviewing the manuscript entitled, “Gamma secretase as a therapeutic target for cardiac disease” by Sen S. et al., the point of view is wonderful, and it has potential as a therapeutic agent for the control of γ-secretase. However, γ-secretase is a versatile enzyme, and the development of drugs targeting γ-secretase is extremely difficult. The authors need to respond to the following concerns.
Concerns
Gamma secretase is a versatile enzyme and there are many substrates in it. Therefore, the treatment for some diseases is effective by suppressing γ-secretase, and the therapy for some diseases are effective by inducing it. The γ-secretase inhibitor is undoubtedly one of the candidates for the treatment of Alzheimer's disease. Although the authors describe that γ-secretase stimulation may be effective in heart disease, as the development of γ-secretase inhibitors is progressing, you need to describe the disadvantages of γ-secretase stimulant administration.
The author should firmly state the definition of cardioprotective. Is this preventive or curative? Furthermore, what do the cells in Figure 2 show? Is this the cardiomyocyte? If it is a cardiomyocyte, the authors need to add the mechanism leading up to the cardioprotective. If not, you need to add a figure with the cardiomyocyte.
Do the authors want to target the γ-secretase itself or the γ-secretase substrates? From the content of this manuscript, the title does not seem to be suitable.
The author needs a detailed explanation of Figure 3. In particular, the authors should describe how advanced the development and research of γ-secretase agonists and antagonists are. This is very important.
From line 172, the authors mentioned anti-fibrotic function of Notch pathway. Does this mechanism focus on cardiac fibroblast or cardiomyocyte? The author should describe in detail the target cells and their mechanisms. And, as mentioned above, you should firmly state the definition of cardioprotective.
Author Response
Reviewing the manuscript entitled, “Gamma secretase as a therapeutic target for cardiac disease” by Sen S. et al., the point of view is wonderful, and it has potential as a therapeutic agent for the control of γ-secretase. However, γ-secretase is a versatile enzyme, and the development of drugs targeting γ-secretase is extremely difficult. The authors need to respond to the following concerns.
Gamma secretase is a versatile enzyme and there are many substrates in it. Therefore, the treatment for some diseases is effective by suppressing γ-secretase, and the therapy for some diseases are effective by inducing it. The γ-secretase inhibitor is undoubtedly one of the candidates for the treatment of Alzheimer's disease. Although the authors describe that γ-secretase stimulation may be effective in heart disease, as the development of γ-secretase inhibitors is progressing, you need to describe the disadvantages of γ-secretase stimulant administration.
Response:
We would like to sincerely thank the reviewer for the feedback. Undoubtedly, we do not know enough yet about what kind of gamma-secretase modulation could benefit cardiac diseases. However, we believe it is important to point out how complicated gamma-secretase modulation could be as a therapeutic approach. To elaborate on this more, we have added more discussion in new section 6 (lines 439-482). Positive gamma-secretase modulators, GSAP and CD147 are associated with undesirable cardiac effects reported in the literature (seen in Table 2). It may be the case that too much gamma-secretase, as well as too little gamma-secretase activity, will not be beneficial to the heart in some settings. Thus, non-selective gamma-secretase stimulation will warrant additional studies to ensure safety in a preclinical setting. Regulating gamma-secretase activity and pushing its activity towards specific targets may be the route forward for treating cardiac disease, a point we elaborated on in our conclusion (lines 505-509, 515-521).
Furthermore, we have added new content to the conclusion (lines 496-509). Here we mention how gamma-secretase stimulation is a risk in the context of Alzheimer’s disease, and how non-selective gamma-secretase activity is linked to Alzheimer’s disease.
The author should firmly state the definition of cardioprotective. Is this preventive or curative? Furthermore, what do the cells in Figure 2 show? Is this the cardiomyocyte? If it is a cardiomyocyte, the authors need to add the mechanism leading up to the cardioprotective. If not, you need to add a figure with the cardiomyocyte.
Response:
We appreciate the concern raised by the reviewer on the use of cardioprotection/cardioprotective throughout the manuscript. Cardioprotection refers to any means that preserves heart function or reduces myocardial loss against injurious conditions. To eliminate any confusion, we now only apply the word “cardioprotection” in pathological conditions that involve massive cell death, such as I/R injury or oxidative stress, which is the most widely accepted in the field.
To further clarify whether a beneficial mechanism is preventive or curative, we formulated a column “Findings suggesting effects in cardiac disease” in Table 1 to aid readers to follow. In this column, we briefly summarized the study design, such as loss-of-function, inhibitor, gain-of-function, etc. and their outcome in cardiac function. We also indicate whether the study is suggesting preventive effect (meaning intervention happened before disease/injury), or curative effect (meaning intervention happened after the onset of disease).
The purpose of Figure 2 is to showcase all known cardiac pathways that are subjected to gamma-secretase regulation, which includes pathways that are reported in non-cardiomyocytes, such as p75NTR. To address reviewer’s concerns on cardiomyocyte-specific mechanisms, we have added more relevant information in Table 1. In the table, we have collated the important findings from the literature that have been mentioned in the paper. The table entails information about the gamma-secretase substrates, their role in the heart, the cell types they function in, and cardiac diseases caused due to dysregulation of these substrates. The table also includes downstream-interacting partners of gamma-secretase substrates like AMPK. We hope that this added information can ease the reviewer's concerns.
Do the authors want to target the γ-secretase itself or the γ-secretase substrates? From the content of this manuscript, the title does not seem to be suitable.
Response:
Thanks to the reviewer for highlighting the need for clarity. Our intention is to focus on gamma-secretase and look at it from the perspective of a therapeutic target for cardiac disease. The function of gamma-secretase is to process substrates and does a great deal of regulation through this processing and the downstream cleaved substrates interactions. So, the goal is to modulate gamma-secretase specifically to affect the level of cleaved substrates, which may allow targeting multiple pathways simultaneously for better therapeutic effects. We have rewritten the second paragraph (lines 39-44) of the introduction, and the final conclusion paragraphs (lines 505-509, 515-521) to better drive this point home.
Furthermore, we have added a section to describe the current findings on cardiac phenotypes of gamma-secretase knockout models (lines 121-142), which can shed some light on the role of gamma-secretase in cardiac physiology. We also included more discussion on the current status of gamma-secretase small molecules with respect to the cardiac implications (Section 6), which makes the review manuscript more gamma-secretase centric.
Lastly, to avoid confusion, we also changed the title to “The Potential of Gamma-Secretase as a Therapeutic Target for Cardiac Diseases”, which will further engage a discussion of the effects of gamma-secretase downstream substrates.
The author needs a detailed explanation of Figure 3. In particular, the authors should describe how advanced the development and research of γ-secretase agonists and antagonists are. This is very important.
Response:
Thanks to the reviewer for the suggestion. To address this, we have added two new sections: 5.2 - Downstream Complexity, and 6 - Difficulty in Drug Development. Section 5.2 discusses the generation and interpretation of Figure 3. We made some changes to figure 3 to better reflect the terminology in section 6 (using acronyms GSMs and GSIs). Section 6 dives into drugs that target gamma-secretase: their classification, roles, effects, and gives a general sense of how developed these drugs are. To summarize, gamma-secretase modulators have been developed extensively in the context of Alzheimer’s disease and cancer. However, all gamma-secretase inhibitor trials have been stopped prematurely. Some other molecules are currently in clinical trials, but the efficacy is in question with considerable side effects. Lastly, some cardiac phenotypes from molecules and proteins that affect gamma-secretase activity are summarized in Table 2. We hope these additions will ease the reviewer’s concerns on the description of current research on gamma-secretase modulators and inhibitors.
From line 172, the authors mentioned anti-fibrotic function of Notch pathway. Does this mechanism focus on cardiac fibroblast or cardiomyocyte? The author should describe in detail the target cells and their mechanisms. And, as mentioned above, you should firmly state the definition of cardioprotective.
Response:
Thank the reviewer for the comment and the opportunity to clarify. We have incorporated the changes to the manuscript as per your comments. We expanded the description of the anti-fibrotic mechanisms by Notch on lines 236-260. Notch partakes in the anti-fibrotic mechanisms in the heart, primarily in the cardiac fibroblasts. The Notch pathway acts antagonistically on cardiac fibrosis. Increased levels of TGF-Beta results in activation of the TGF-Beta/Smad pathway that increases the conversion of cardiac fibroblasts to myofibroblasts, a hallmark of cardiac fibrosis. The myofibroblasts contribute to increased secretion and deposition of the extracellular matrix as well as collagen type I that disrupts the myocardial architecture and establishes cardiac fibrosis. The conversion of fibroblast to myofibroblast was accompanied by a decrease in Notch1, 3, and 4 expressions. TGF-beta/Smad3 pathway is the primary pathway responsible for promoting the activation of cells and subsequent fibrogenesis. The Notch pathway helps prevent fibrosis by inhibiting TGF-beta and Smad3 pathways via Notch 3. On the other hand, Notch inhibition has been shown to upregulate cardiomyocyte apoptosis as well as hypertrophy in cardiac cells. Thus, it provides additional environmental cues for the initiation of fibrosis as a compensatory process to cardiac damage. However, in the long term, fibrosis will contribute to cardiac dysfunction due to irreplaceable scar formation. By inhibiting both apoptosis of cardiomyocyte cells and the fibrotic process, cardiac function can be preserved, which is indicated in a study by Kratsios et al, 2010. We also included the short description and reference in Table 1.
Reviewer 4 Report
REVISEAuthors presented an updated overview of the current knowledge of gamma-secretase processing of cardiac-relevant substrates and seeks to understand if modulation of gamma-secretase activity would be beneficial to combat cardiac disease.
- The paper is quite well-written and interested.
- Authors should prepare a Figure to summarize the role of gamma-secretase in the specific cell, i.e. cardiomyocyte or at least prepare a Table to summarize which disease can be associated with gamma-secretase abnormalities and with which pathway.
- References should be formatted in one style.
- I think that authors should also include some most recent studies such as:
- - Kim K, Yu J, Kang JK, Morrow JP, Pajvani UB. Liver-selective γ-secretase inhibition ameliorates diet-induced hepatic steatosis, dyslipidemia and atherosclerosis. Biochem Biophys Res Commun. 2020 Jul 5;527(4):979-984.
- - Yu Q, Kou W, Xu X, Zhou S, Luan P, Xu X, Li H, Zhuang J, Wang J, Zhao Y, Xu Y, Peng W. FNDC5/Irisin inhibits pathological cardiac hypertrophy. Clin Sci (Lond). 2019 Mar 1;133(5):611-627.
- - Rivera-Torres J, Guzmán-Martínez G, Villa-Bellosta R, Orbe J, González-Gómez C, Serrano M, Díez J, Andrés V, Maraver A. Targeting γ-secretases protect against angiotensin II-induced cardiac hypertrophy. J Hypertens. 2015 Apr;33(4):843-50
- - Bovo E, Nikolaienko R, Kahn D, Cho E, Robia SL, Zima AV. Presenilin 1 is a direct regulator of the cardiac sarco/endoplasmic reticulum calcium pump. Cell Calcium. 2021 Nov;99:102468.
Author Response
Authors presented an updated overview of the current knowledge of gamma-secretase processing of cardiac-relevant substrates and seeks to understand if modulation of gamma-secretase activity would be beneficial to combat cardiac disease.
The paper is quite well-written and interested.
Authors should prepare a Figure to summarize the role of gamma-secretase in the specific cell, i.e. cardiomyocyte or at least prepare a Table to summarize which disease can be associated with gamma-secretase abnormalities and with which pathway.
Response:
A huge thank to the reviewer for the encouraging words and thoughtful insights.
While figure 2 was created to include all pathways identified in cardiac-related diseases, certain proteins, such as p75NTR, are expressed in other cardiac cells, but not in cardiomyocytes. To address the reviewer’s concern, we made Table 1 to clarify cell-specific pathway alterations and their associated abnormalities. In the table, we have collated all the important findings from the literature that has been mentioned in the paper. The table entails information about the gamma-secretase substrates, their roles in the heart, the corresponding cell types as well as cardiac diseases caused due to dysregulation of these substrates. We also included the speculation of the overall protective and detrimental effects that could be achieved on manipulating gamma-secretase activity to modulate the activity of these substrates. The table also includes downstream-interacting partners of gamma-secretase substrates like AMPK. In addition, we also made Table 2 to summarize the cardiac phenotypes with gamma-secretase abnormalities induced by genetic ablation in mice. We hope these additions will ease the reviewer’s concern.
References should be formatted in one style.
Response:
Thanks to the reviewer for the feedback. We are dealing with some technical issues regarding our references, which have been compiled in LaTeX. We have been assured from the MDPI LaTeX support that the production team will take care of this.
I think that authors should also include some most recent studies such as:
- Kim K, Yu J, Kang JK, Morrow JP, Pajvani UB. Liver-selective γ-secretase inhibition ameliorates diet-induced hepatic steatosis, dyslipidemia and atherosclerosis. Biochem Biophys Res Commun. 2020 Jul 5;527(4):979-984.
- Yu Q, Kou W, Xu X, Zhou S, Luan P, Xu X, Li H, Zhuang J, Wang J, Zhao Y, Xu Y, Peng W. FNDC5/Irisin inhibits pathological cardiac hypertrophy. Clin Sci (Lond). 2019 Mar 1;133(5):611-627.
- Rivera-Torres J, Guzmán-Martínez G, Villa-Bellosta R, Orbe J, González-Gómez C, Serrano M, Díez J, Andrés V, Maraver A. Targeting γ-secretases protect against angiotensin II-induced cardiac hypertrophy. J Hypertens. 2015 Apr;33(4):843-50
- Bovo E, Nikolaienko R, Kahn D, Cho E, Robia SL, Zima AV. Presenilin 1 is a direct regulator of the cardiac sarco/endoplasmic reticulum calcium pump. Cell Calcium. 2021 Nov;99:102468.
Response:
Thanks to the reviewer for suggesting these recent papers. As per your suggestion, we have incorporated three of the suggested studies into our manuscript, which we felt added greatly to our paper in lines: 458, 490, and 130 (in the order listed above). However, we want to limit the scope of our paper to gamma-secretase and modulators as a therapeutic in the field of cardiac diseases. So, we did not include the study by Yu et al. in our manuscript because gamma-secretase was not an upstream activator in the study. Also, the interplay of AMPK and mTOR does not fit in our current manuscript context.